# Use of High-Plex Data Reveals Novel Insights into the Tumour Microenvironment of Clear Cell Renal Cell Carcinoma

**DOI:** 10.3390/cancers14215387

**Published:** 2022-11-01

**Authors:** Raffaele De Filippis, Georg Wölflein, In Hwa Um, Peter D. Caie, Sarah Warren, Andrew White, Elizabeth Suen, Emily To, Ognjen Arandjelović, David J. Harrison

**Affiliations:** 1School of Medicine, University of St Andrews, St Andrews KY16 9TF, UK; 2School of Computer Science, University of St Andrews, St Andrews KY16 9SX, UK; 3Indica Labs, Albuquerque, NM 87114, USA; 4NanoString Technologies, Seattle, WA 98109, USA

**Keywords:** multiplex, immunofluorescence, nanostring, image analysis, pathology, kidney, spatial analysis

## Abstract

**Simple Summary:**

Cancer is a complex ensemble of morphological and molecular features whose role is still unclear. Moreover, their role may change in different areas of the same tumour. Artificial intelligence (AI) allows pathologists to go beyond human perception and bias and may help better understand how these features influence disease progression. Furthermore, by capturing variation intrinsic to the tumour, AI may improve the accuracy of current prognostic tools, such as Leibovich Score (LS), in predicting patient outcome and response to therapy. For these reasons, we studied clear cell renal cell carcinoma (ccRCC) tissue in which molecular features and their coexpression in the same cell were quantified and mapped using AI-based image analysis software. We demonstrated a novel approach for investigating ccRCC and revealed new potential biomarkers of prognosis which may also be able to direct patients towards the most appropriate personalised therapy.

**Abstract:**

Although immune checkpoint inhibitors (ICIs) have significantly improved the oncological outcomes, about one-third of patients affected by clear cell renal cell carcinoma (ccRCC) still experience recurrence. Current prognostic algorithms, such as the Leibovich score (LS), rely on morphological features manually assessed by pathologists and are therefore subject to bias. Moreover, these tools do not consider the heterogeneous molecular milieu present in the Tumour Microenvironment (TME), which may have prognostic value. We systematically developed a semi-automated method to investigate 62 markers and their combinations in 150 primary ccRCCs using Multiplex Immunofluorescence (mIF), NanoString GeoMx^®^ Digital Spatial Profiling (DSP) and Artificial Intelligence (AI)-assisted image analysis in order to find novel prognostic signatures and investigate their spatial relationship. We found that coexpression of cancer stem cell (CSC) and epithelial-to-mesenchymal transition (EMT) markers such as OCT4 and ZEB1 are indicative of poor outcome. OCT4 and the immune markers CD8, CD34, and CD163 significantly stratified patients at intermediate LS. Furthermore, augmenting the LS with OCT4 and CD34 improved patient stratification by outcome. Our results support the hypothesis that combining molecular markers has prognostic value and can be integrated with morphological features to improve risk stratification and personalised therapy. To conclude, GeoMx^®^ DSP and AI image analysis are complementary tools providing high multiplexing capability required to investigate the TME of ccRCC, while reducing observer bias.

## 1. Introduction

Clear cell renal cell carcinoma (ccRCC) is the most frequent kidney cancer and the deadliest disease of the urinary tract. Over 10,000 new cases are diagnosed annually, with over 76,000 new renal cell carcinoma (RCC) cases diagnosed in 2021 [1]. To date, the only curative method is nephrectomy, although 30% of patients experience recurrence [2]. Even though tremendous progress has been made in systemic therapy with the advent of immune checkpoint inhibitors (ICIs), around one-third of ccRCC patients present with metastasis, and their 5-year survival is only 12% [3,4].

The current clinical gold standard prognostic tools are TNM staging and International Society of Urological Pathology (ISUP) nuclear grading, obtained through Haematoxylin & Eosin (H&E) staining of resected tissue sections. The former considers the tumour size (pT), lymph node involvement (pN), and the presence of distant metastasis (pM); the latter takes into account nucleolar prominence and morphology, as well as the presence of rhabdoid and sarcomatoid features [5,6].

Several prognostic algorithms have been built to assign prognostic scores to patients and predict the risk of advanced disease. For instance, the widely used Leibovich score (LS) predicts the risk of recurrence after surgery by considering pT, pN, tumour size, nuclear grade, and the presence of necrosis [7] (see Appendix A Table A1). However, LS only considers tumour morphological features, which are subjectively assessed by pathologists, while there is a lack of general consensus about molecular features. For these reasons, current prognostic tools, such as LS, are subject to inter-observer bias and are unable to accurately predict patient outcome and response to therapy [6,8].

Other factors may influence tumour progression: ccRCC is characterised by a highly heterogeneous tumour microenvironment (TME), in which hypoxia, blood vessels, stromal, and immune cells play a crucial role for cancer progression. For example, variations in immune escape mechanisms, such as PD-1/PD-L1 interaction, which are at the base of some ICIs, are responsible for drug resistance [9]. Therefore, assessing molecular marker expression such as PD-1 and PD-L1 could predict the response to therapy and distinguish patients who would benefit from treatment from those who would potentially experience adverse effects. After exposure to chronic inflammation, cytotoxic T cells express high levels of co-inhibitory receptors, including PD-1, LAG-3, TIM-3, and CTLA-4. This results in T cell loss of function, and tumour adaptation to the immune response, favouring disease progression. The main exhausted T cell molecular signature is represented by coexpression of CD8 along with PD-1 and TIM-3 and was associated to poor prognosis in ccRCC [10].

Moreover, recent evidence supports the hypothesis that high PD-1/PD-L1 expression and increased regulatory T cell infiltration promote epithelial-mesenchymal transition (EMT), a process in which epithelial cells acquire a mesenchymal phenotype [11]. These signals lead to cytoskeleton reorganisation and cell motility, which are associated with invasiveness and metastatic potential in cancer [12]. The hallmark of EMT is the upregulation of N-cadherin followed by the downregulation of E-cadherin [13]. Although the origin of EMT is still unclear, evidence suggests that the transforming growth factor β (TGF-β) induces EMT by activating transcription factors including vimentin, snail, and E-box binding homeobox 1 (ZEB1), which suppress epithelial cell proliferation [14,15,16]. In particular, vimentin is involved in the cytoskeleton reorganisation [17], whereas ZEB1 represses E-cadherin expression and activates N-cadherin and vimentin expression [18].

Interestingly, it has recently been shown that cancer stem cells (CSCs) require a hybrid epithelial/mesenchymal phenotype in order to initiate cancer progression [19]. These de-differentiated tumour cells hold self-renewal potential and cell motility which contribute to tumour aggressiveness and poor prognosis [20].

However, no standard CSC markers exist for ccRCC due to the high heterogeneity of its TME, although some have shown a prognostic potential. For instance, CD44 is involved in the regulation of stem cell features via the Wnt/β-catenin signalling pathway, suggesting that this marker could contribute to EMT [21]. Moreover, CD44 was associated with poor 5-year survival, high nuclear grade, and recurrence in ccRCC [22]. To note, in vitro studies using RCC cell lines observed that CD44 was upregulated in the presence of tumour-associated macrophages (TAMs) [23].

OCT4 has also been shown to be responsible for stemness, EMT, tumorigenesis, and cell survival mechanisms [24]. In ccRCC, OCT4a, either alone or in combination with NANOG, was associated with stemness and poor prognosis [25]. CSCs have also been shown to bind to CTLA-4 receptor causing CTLA-4 upregulation. This evidence suggests that CSCs may also be involved in immune evasion and T cell exhaustion [26,27], showing how all the mechanisms described above are strictly connected to each other symbiotically modulating the TME.

Understanding how the high ccRCC intra-tumour heterogeneity (ITH) affects tumour progression and drug resistance is crucial for stratifying patients’ prognosis and personalising the therapy. Therefore, a better investigation of ccRCC TME is necessary, and novel approaches, such as multiplex immunofluorescence (mIF) and spatial profiling in situ can play a crucial role. Artificial intelligence (AI) and machine learning (ML) applied to image analysis have revolutionised digital pathology, allowing more precise quantification, generating more data, and overcoming inter-observer variability [28].

We systematically developed a semi-automated method to assess 62 and 158 features in Tissue Microarray (TMA) and whole slide tissue sections, respectively. Subsequently, we assessed their relationship and their correlation with outcome using ML-based statistical tools (Figure 1). This approach may facilitate the discovery of novel molecular signatures able to predict outcome, choose the best therapeutic strategy, and avoid ineffective and potentially harmful drug administration.

## 2. Materials and Methods

### 2.1. Study Population

Tissue samples resected from 150 patients who were diagnosed with ccRCC and treated by radical nephrectomy between the years 1983 and 2018 were extracted from the pathology archive in Edinburgh and verified by an experienced pathologist. Specimens were collected prior to therapy administration. None of these patients had therapy before surgery, and none received immune blockade therapy.

The analysed cohort consisted of 60 females and 86 males. The gender of 4 patients was not available. The average age was 64 years, and mean follow-up was 60 months. Some 106 patients (70%) were reported to be deceased, of which 75 (50%) experienced cancer-related death (CRD). The 5-year survival rate was 66%, 46 patients (30%) experienced metastasis at the time of diagnosis and for them, overall survival (OS) rate was 46%, and 5-year survival rate was 58%.

A total of 124 patients had LS data, although no necrosis data were available. The majority of patients presented stage 3 and intermediate LS risk, while nuclear grades 2 and 3 were the most common. Finally, a total of 18 patients (14.5%) presented lymph node involvement (pN = 1), and 35 patients (28.2%) presented distant metastasis. More detailed information is available in Appendix B Table A2.

From each primary sample, 3 tissue punches were randomly collected and stored in 11 TMA blocks. Clinical information was retrieved from the patients’ medical records, including age, sex, cancer-specific survival, time after initial surgery, TNM stage, and nuclear grade. The tissue samples, clinical data, and ethical approval for this retrospective study were issued by the Research Tissue Bank and the SAHSC Bio Resource on behalf of NHS Lothian (15/ES/0094).

### 2.2. Antibody Optimisation

Antibodies were first optimised separately on optimisation TMAs, consisting of several different normal and cancer tissues (RCC, breast, gastric, colon, prostate, tonsil, spleen, etc.) and then on RCC TMAs from the study cohort. All antibodies were optimised first in brightfield immunoperoxidase and then for immunofluorescence (IF). For tumour cell segmentation in IF, a combination of CA9 and pan-Cadherin antibodies was applied. The list of antibodies used, along with the dilution used is shown in Table A3 in Appendix C.

### 2.3. Multiplex Immunofluorescence

Formalin-fixed, paraffin-embedded (FFPE) tissues were first de-waxed in xylene and then re-hydrated through descending concentrations of ethanol. In order to expose epitopes, heat-mediated antigen retrieval was performed in an electric pressure cooker using TRIS-EDTA buffer (pH 9). Subsequently, slides were treated with 3% hydrogen peroxide to block endogenous peroxidase activity, and with protein block (Agilent, Santa Clara, CA, USA, X090930-2), in order to prevent non-specific staining. Slides were then incubated with primary antibodies for either one hour or 30 min at room temperature or overnight at 4 °C. After primary incubation, tissues were then incubated with secondary HRP-conjugated or biotin-conjugated antibodies for 30 min at room temperature. Slides were treated either with cyanine 3, cyanine 5, fluorescein, using Tyramide Signal Amplification (TSA) system, or Alexa Fluor 750 (ThermoFisher, Waltham, MA, USA, S21384), diluted in amplification diluent (Perkin Elmer, FP1498) and antibody diluent (Agilent, S080983-2), respectively. Hoechst 33342 (ThermoFisher, H3570), diluted in PBS, was used for nuclear counterstaining. Finally, slides were dehydrated in ethanol, air-dried, and mounted using Prolong anti-fade mounting medium (ThermoFisher, P36930). The full list of primary antibodies used is shown in Table A3 of Appendix C.

### 2.4. GeoMx^®^ Digital Spatial Profiling

#### 2.4.1. Sample Preparation

Slides were prepared according to the digital spatial profiling (DSP) FFPE Protein Manual (MAN-10100-05). FFPE slides were baked for one hour at 60 °C before being treated with CitroSolv (Fisher, 22-143-975) and descending concentrations of ethanol. Antigen retrieval in citrate buffer (pH 6) was performed in a pressure cooker and slides were then washed in TBS-T, blocked with Buffer W (Iba 2-1003-100) and incubated overnight at 4 °C with 62 UV-cleavable oligonucleotide-conjugated antibodies for DSP and fluorescent-labelled antibodies for visualisation (Pan Cadherin–Alexa Fluor 488, Syto83-Alexa Fluor 532, CD3e-Alexa Fluor 594, CD163-Alexa Fluor 647). The oligonucleotide-conjugated antibodies consisted of five panels (immune cell profiling, immune activation status, immune cell typing, tumour markers, and drug targets), and are shown in Table A4 of Appendix C.

#### 2.4.2. DSP and nCounter^®^ Readout

After incubation, the slides were loaded into the GeoMx^®^ digital spatial profiler, where the tissues were scanned and digital images were generated for region of interest (ROI) selection. Each TMA core corresponded to an ROI. Subsequently, ROIs were illuminated with UV light, and oligo tags were released into the aqueous layer just above the slide, which was aspirated through a micro-capillary tube and stored in a single well of a 96-well plate. This process was repeated for each ROI. An average of 120 ROIs per slide were collected, and nCounter Readout was processed according to the GeoMx DSP Readout Manual (MAN-10091-09), and as previously described [29,30].

#### 2.4.3. DSP Quality Control and Normalisation

Quality control was based on marker density and was automatically performed by the nSolver^®^ software by NanoString Technology. Data normalisation was based on mouse IgG_1_ and IgG_2_ antibodies included in the antibody mix, which were considered as negative controls. Raw data were converted to a $\log_{2}$ scale. For each ROI, corresponding to the TMA core, the mean of the IgG_1_ and IgG_2_ was used to calculate the signal to noise ratio (SNR) as
(1)$$\mathrm{SNR}=X-\frac{{\mathrm{IgG}}_{1}+{\mathrm{IgG}}_{2}}{2}$$
where *X* is the count of a single marker in a single ROI. Since NanoString suggests that SNR should be >$\log_{2}\left(3\right)$, values greater than 1.5 were considered a positive signal. In order to keep the data intact, values below 1.5 were clamped to 1.5.

### 2.5. Image Analysis

TMA images were analysed using Definiens^®^ Tissue Studio. Around 10% of images were used for training, and several training rounds were performed to reach optimal results. For each TMA core, tumour-stroma segmentation was performed, and only the tumour areas were considered since most cores consisted of over 90% tumour (Figure 2A,B). Nuclear segmentation was based on the Hoechst channel. Subsequently, cells were reconstructed virtually. In the case of tumour cells, the expansion followed the tumour mask stain (Figure 2C). In the case of immune cells, the nuclear circumference was expanded by a specific radius. Whole slide images were analysed with Indica Labs Halo^®^ AI software. Like with Tissue Studio, only tumour regions were considered after tumour-stroma segmentation, and nuclear segmentation was based on the Hoechst intensity. After segmentation, Halo^®^ AI was fed with hundreds of examples in order to distinguish three nuclear types: tumour, immune, and stromal nuclei (Figure 2D). Cell simulation was only performed by expanding nuclear circumference since no tumour mask was used, and tumour cells were recognised from their nuclear morphology. Subsequently, cells were classified accordingly, and single/multiple expressions were quantified.

Spatial analysis was also performed for whole slides using dHalo^®^ software. Four different analyses were performed: infiltration analysis to determine the density of PD-1^+^ T cells in 5 distance bands within and outside the tumour margin (Figure 3); nearest neighbour analysis to determine the number of PD-1^+^ T cells nearby each PD-L1^+^ tumour cell within a specific radius; proximity analysis to determine the distance of PD-1^+^ T cellos from PD-L1^+^ tumour cells; and density heatmap analysis to visualise the density of PD-L1 tumour cell density across the tissue section (Figure 4).

### 2.6. Statistical Analysis

Statistical analysis of 67 and 152 molecular features was performed in TMAs and WSs, respectively, using R Studio (version 1.1.423) and Python (version 3.8). Seven morphological features, including maximum diameter, pN, stage, Fuhrman grade, ISUP grade, classical LS, and ISUP LS were also investigated in both experiments. Unsupervised clustering according to Pearson’s correlation was used to investigate relationships between markers. Cox’s proportional hazard method was used to predict 5-year and OS, where CRD was considered as the event. OS and CRD data were collected from the Research Tissue Bank of the Royal Infirmary of Edinburgh. We conducted two sets of experiments: one where we took as input the continuous data and another where each variable was binarised into *high* and *low* after determining the optimal cutpoint in relation with outcome, using the maximally selected rank statistics from the R package *maxstat*. Statistical significance was evaluated using the likelihood ratio test and log-rank test to obtain *p*-values for the continuous and categorical models, respectively. We exhaustively fit Cox models for every combination of one, two, and three markers, and applied Chow–Denning correction [31] to the *p*-values in order to account for multiple testing. Due to the vast number of models that needed to be fit (in the order of millions), we ran the computation in parallel on 20 Linux machines for approximately 12 h using the Jug library [32] to parallelise the code.

## 3. Results (Tissue Microarray)

For each molecular marker, density was calculated by dividing the number of positive cells by the TMA core area. Subsequently, average density of replicates was calculated.

### 3.1. Pearson’s Correlation

Unsupervised clustering using Pearson’s correlation found positive correlation among morphological, DSP-detected, and IF-detected markers, involved in immune response, immune evasion, T cell exhaustion, as well as CSCs and EMT (Figure 5).

Among DSP markers, positive correlation was found between CD163 and CD4 and CD44 and CD68. CD3 was positively correlated with CD4, CD44, CD68, and CD8. Moreover, CD44 was positively correlated with CD68. CD8 was positively correlated with CD4 and CD68 (Figure 5).

Among IF markers, β-catenin was positively correlated with DSP Histone H3. PD1^+^CD8^+^ cells were positively correlated with CD8^+^LAG-3^+^ cells and with CD8^+^LAG-3^+^TIM-3^+^ cells. Positive correlation was also found between ZEB1 and TIM-3, as well as CD8 and the following markers: PD-1, PD-L1^+^ tumour cells, CD163, and LAG-3. Interestingly, positive correlation was also found between CD8^+^LAG-3^+^ cells and CD8^+^PD-1^+^ cells (Figure 5), indicating that PD-1 may contribute to an exhausted T cell phenotype.

### 3.2. Survival Analysis

#### 3.2.1. Univariate Cox Regression

Overall, morphological features showed the highest significance. In particular, the LS with the ISUP nuclear grade instead of the Fuhrman grade was the most significant feature and was associated with poor OS, (see Figure 6) and 5-year survival, (see Figure 6B). Similar results were obtained with pN, Fuhrman LS, and ISUP grade. Among molecular markers, OCT4 alone and in combination with ZEB1 was associated with poorer outcome. Moreover, OCT4^+^ZEB1^+^β-catenin was associated with 5-year survival.

CD8, total tumour cell density, and PD-L1^−^ tumour cell density were associated with longer OS and 5-year survival. CD34 density was associated with longer OS, while DSP-detected VISTA was associated with longer 5-year survival.

#### 3.2.2. Multivariate Cox Regression

We exhaustively fit multivariate Cox regression models on every combination of two and three features in order to find combinations of markers that exhibit statistically significant predictions of patient outcomes. For the TMA cohort with 74 features, we consequently fit 742+743+744 = 1,218,151 Cox models. After Chow–Denning correction, we found numerous groups of markers that were predictive for OS or 5-year survival. As expected, most of these groups included features which individually were already statistically significant in the univariate model. However, some groups comprised entirely of markers that were not statistically significant when considering the markers individually in univariate Cox regression, but in combination achieved statistical significance.

In particular, coexpression of CD8 and LAG-3, representing an exhausted T cell phenotype, along with HLA DR, an isotype of the human leukocyte antigens, was identified as a prognostic combination for 5-year survival, suggesting that MHC II plays a crucial role in T cell exhaustion. OCT4 alone or coexpressed with vimentin, reached statistical significance for OS and 5-year survival, respectively, after pairing with Arginase 1 (ARG1), suggesting that CSC and EMT may be linked, favouring cancer proliferation. OCT4 combination with HLA DR was also associated with shorter 5-year survival. OCT4^+^vimentin^+^ cancer cells also stratified patients by OS when combined with PD-1 expression and snail. Unexpectedly, PD-1^+^ cell expression was associated with longer OS when combined with snail^+^vimentin^+^ tumour cells, therefore further investigation is needed. When OCT4^+^β-catenin^+^ cancer cell expression was combined with TIM-3, detected through DSP, it showed shorter OS and 5-year survival. Instead, OCT4^+^ZEB1^+^β-catenin^+^ tumour cell expression showed shorter OS when combined with snail^+^vimentin^+^ tumour cell expression. This set of marker groups may provide insight into complex interactions between features that have an impact on predicting patient outcome. Table 1 lists all pairs of markers (i.e. combinations of two) in this category. We mention some statistically significant combinations of more than two markers in this paper but do not provide the full list for the sake of brevity.

As is evident in Figure 6A, Cox regression showed that OCT4^+^ZEB1^+^ tumour cells were associated with CRD. However, tumour cells only expressing OCT4 were not found to predict the outcome with statistical significance and neither did the ones only expressing ZEB1. Furthermore, in bivariate Cox regression, OCT4^+^ and ZEB1^+^ in combination was not a statistically significant predictor of outcome. This means that merely knowing the densities of OCT4^+^ cells and ZEB1^+^ cells is insufficient; instead, knowledge of the density of cells coexpressing both markers is required to arrive at a prediction carrying statistical significance. This highlights the need for accurate and precise spatial analysis that can identify co-registered markers on the same cell. This result also suggests that coexpression of OCT4 and ZEB1 in tumour cells may not only be indicative of a dedifferentiation, but also that CSC and EMT may be dependent mechanisms.

### 3.3. Integrated LS

While the LS is able to stratify patients into low, intermediate, and high risk groups, patients at intermediate risk still exhibit varied outcomes. In this section, we evaluate how a combination of the LS with molecular markers can result in better stratification.

As baseline, we assessed the stratification ability of the traditional LS on the TMA cohort. After scoring the patients, a Kaplan–Meier analysis was able to accurately separate high and intermediate risk groups (p<0.0001) (Figure 7). Only six patients showed low risk score (0–3) and were therefore excluded from the analysis.

Since traditional LS only relies on morphological data, we tested the stratification ability of the molecular markers that resulted in significant survival analysis. In particular, we found that two specific molecular markers were able to further stratify the patients at intermediate risk according to ISUP LS: IF-detected OCT4^+^ tumour cell density was used to stratify patients into intermediate–high and intermediate–low groups (Figure 8A). Eight patients were excluded from this analysis since their OCT4 data were not available. Similarly, CD34, detected through DSP, showed the same stratification ability (Figure 8B).

To further demonstrate the prognostic value of these markers, they were integrated in the LS algorithm. The patients were categorised in high and low risk according to marker expression and LS. Interestingly, OCT4^high^ integrated in the ISUP LS was able to significantly stratify patients (Figure 9A). The same result was obtained when CD34^high^ was integrated into the algorithm (Figure 9B).

## 4. Results (Whole Slides)

### 4.1. Pearson’s Correlation

Positive correlation was found between CD8^+^ immune cells and PD-L1^+^ tumour cells and between OCT4a^+^ tumour cells and ZEB1^+^SNAIL^+^ tumour cells. Interestingly, a positive correlation was also found between OCT4a^+^ tumour cells and mean tumour cell nuclear area, although no correlation was found with tumour grade. Positive correlation was found between PD1^+^CD8^+^ T cells and PD-L1^+^ tumour cells. TIM-3^+^PD-L1^+^ tumour cells were positively correlated with PD-1^+^CD8^+^ T cells. A heatmap of pairwise correlations is provided in Figure 10.

### 4.2. Cox Regression

Categorical univariate Cox regression showed that ZEB1^+^snail^+^ tumour cells and snail^+^CD44^+^ tumour cells were associated with longer OS and 5-year survival. Unexpectedly, snail^+^ZEB1^+^CD44^+^ tumour cell density was associated with longer 5-year survival (Figure 11). Therefore, further analysis is needed in order to better reveal the role of these markers.

Multivariate analysis showed that a combination of PD1^+^ T cells and ZEB1^+^snail^+^ predicted 5-year survival, whereas these two features did not reach statistical significance alone.

Survival analysis was performed to predict OS and 5-year survival using the classic ISUP and integrated LSs. The classic LS’s stratification ability in the WS cohort is shown in Figure 12.

However, stratification of patients at intermediate risk did not show significant results due to the small sample size.

## 5. Discussion

Although in recent years systems pathology has made great steps, it still lacks standardised methods with high-plex capability, which are necessary to investigate the complex tumour microenvironment. Nanostring GeoMx^®^ DSP overcomes this problem by allowing a high-plex, spatially-resolved assessment of FFPE sections while retaining tissue integrity [33]. In our work, this high-throughput method allowed the identification of positive correlations between a large number of markers. In particular, the CSC marker CD44 was positively associated with immune markers, such as CD163, indicating a role of M2 macrophages in tumour dedifferentiation, or CD3 and CD4, which are indicative of an immune response. These findings support the hypothesis of cross talk between CSCs and TAMs, where TAMs exert a trophic function on stem cells while, in turn, CSCs drive M2 macrophage polarisation [34]. This mechanism may represent an innovative therapeutic target.

Combining GeoMx^®^ DSP with mIF is a novel strategy to select specific regions of interest (ROIs) and obtain more precise information about the role that each marker plays in different areas of the tumour. In this study, GeoMx^®^ DSP was performed using TMAs to allow assessment of a greater number of patients and acquire more robust data. In this setting, each ROI corresponded to a single TMA core. This allowed assessment of co-localisation and hence the relationship among markers. In multivariate analysis, OCT4 and ZEB1 co-expression was only significant for prognosis when the two molecules co-expressed in the same cell. When detected separately, the markers (that is, without co-localisation information) and used as two separate input features in a multivariate model did not reach statistical significance.

ZEB1 is a transcription factor that promotes epithelial-to-mesenchymal transition (EMT) by downregulating the epithelial marker E-cadherin, therefore facilitating cell motility and cancer dissemination [35]. In whole slides, ZEB1 correlated with longer overall survival (OS) when co-expressed in the same cell with snail, and longer 5-year survival when co-expressed with CD44 and snail, suggesting that ZEB1 is not the only determinant of outcome and that other factors may prevent EMT in these patients. By contrast, in TMAs, levels of ZEB1 were associated with shorter OS and 5-year survival when co-expressed with OCT4, consistent with the possibility that OCT4 and ZEB1 both play a role in the acquisition of either a CSC or EMT phenotype [36]. Further investigation is required.

The difference between whole slide and TMA may be because whole slide incorporates tumour core and invasive margin, whereas TMA cores were usually sampled from tumour core. ZEB1, snail, CD44, and OCT4 have been reported in the literature to be involved in tumour de-differentiation [37], therefore it is plausible that this mechanism occurs in the tumour core, which is more often hypoxic and hosts stem cell niches [38].

We have shown a pipeline of work that can be used to interrogate samples of cancer from patients using several analytical approaches. AI-based image analysis of both mIF and DSP produces an overwhelming amount of data which was possible to analyse only with machine learning-based statistical methods and required computational capability alongside substantial engineering efforts in terms of parallel computing to exhaustively assess all feature combinations. When analysing large volumes of data, the risk of overfitting is proportional to the number of features analysed, and further increases when the number of patients is not large enough [39]. We applied Chow–Denning correction [31] to account for multiple testing, which is slightly less conservative than Bonferroni correction [40]. However, further validation with an external cohort is required to confirm our findings.

It is noteworthy that samples were taken before therapy administration, and no therapy information was available in our metadata, so different therapeutic strategies may have been applied to the patients, therefore differently influencing outcome. This is important to consider also due to the large time span between the first and last sample collection, which may also be influenced by the evolution of operative techniques in sample collection. Another aspect to take into account is the deterioration of the FFPE tissue quality, since it has been reported that antigen quality may decay with time, especially those expressed in the nucleus or on the cell surface [41]. However, we have conducted many studies using quantitative mIF, and this has not appeared to be a problem. Moreover, mRNA extraction from similar old FFPE samples was previously validated [42], while no significant mRNA deterioration was found when FFPE slides at different storage time were compared after DSP analysis [43].

To conclude, both statistical analyses in whole slide and TMA samples found that morphological features on their own were stronger predictors than the molecular ones. An AI-based approach to better profile morphology of ccRCC TME could help standardise prognostic tools and overcome inter-observer bias. However, when molecular markers were integrated in the LS, the accuracy in stratifying patients at intermediate risk significantly improved. Therefore, molecular markers are crucial for identifying differences that are not noticeable by only looking at the tumour morphology. Moreover, adding prognostic immune markers to these scoring algorithms may add important information about ICI response and may be crucial to apply the appropriate personalised therapy.

## Figures and Tables

**Figure 1 cancers-14-05387-f001:**
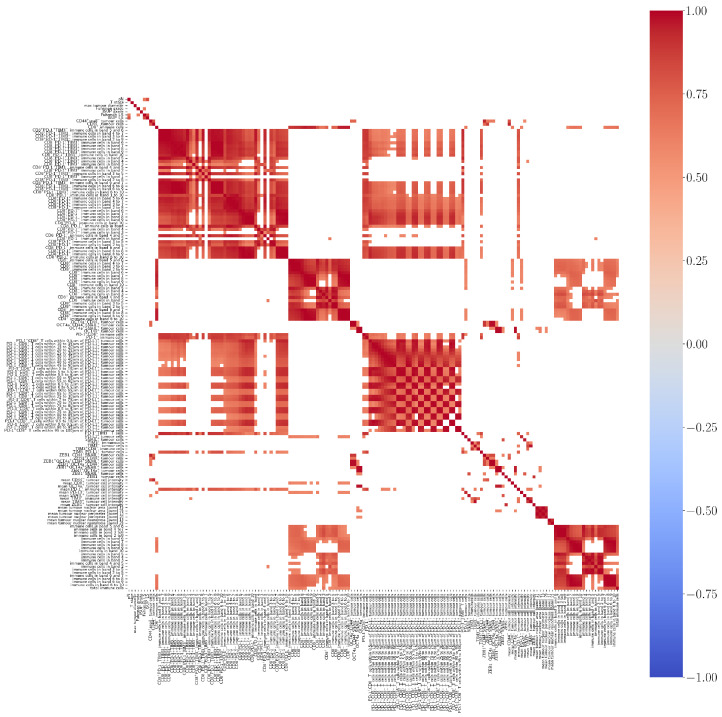
Heatmap showing positive (red), negative (blue), or no (white) correlation among markers assessed in the whole slides (WSs) after unsupervised clustering. A total of 156 features were assessed, including morphological features, molecular marker combination in tumour and immune cells, and spatial analysis data. Only correlations with p<0.05 after Chow–Denning correction are shown. No negative correlations were reported.

**Figure 2 cancers-14-05387-f002:**
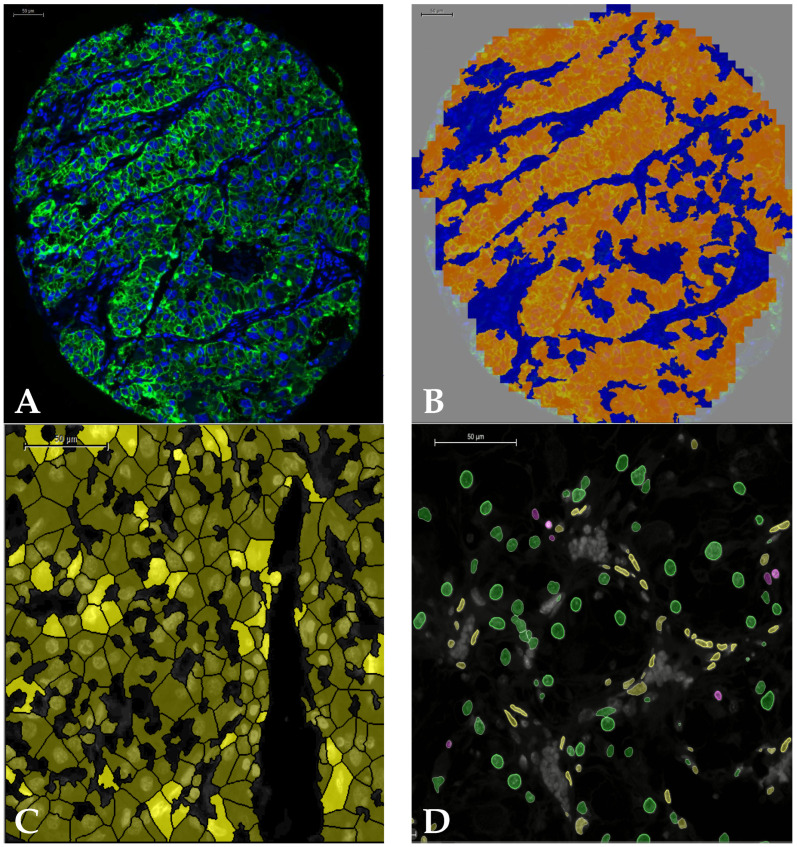
(**A**) TMA core showing nuclear (blue) and tumour (green) stain; (**B**) same TMA core after tumour (orange) stroma (blue) segmentation in Definiens^®^ Tissue Studio; (**C**) tumour cell simulation following tumour mask: positive tumour cells are shown in dark yellow, while tumour cells positive for PD-L1 are shown in bright yellow; (**D**) nuclear phenotyping only based on nuclear morphology using Halo^®^ AI showing tumour nuclei (green), immune nuclei (purple), and fibroblast nuclei (yellow). Red blood cells (gray) were excluded from the analysis.

**Figure 3 cancers-14-05387-f003:**
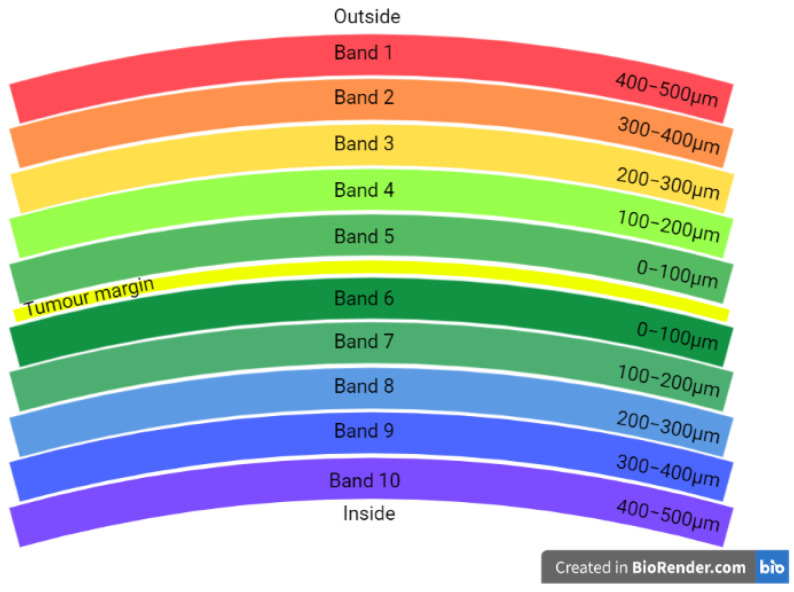
Schematic representation of distance bands assessed in infiltration analysis. The tumour margin is represented by the yellow line between band 5 and 6. Each band was 100 μm in thickness. Marker densities were quantified in each band separately as well as in multiple adjacent bands (see Figure 1).

**Figure 4 cancers-14-05387-f004:**
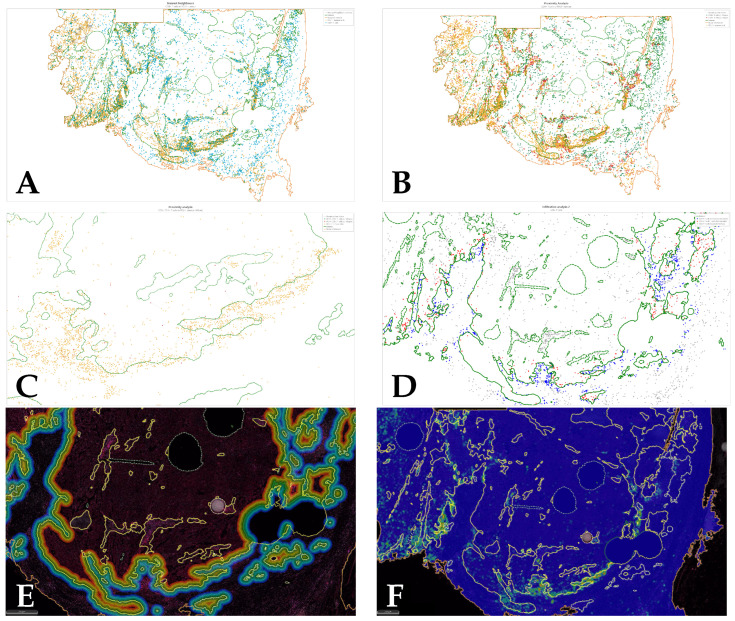
(**A**) Nearest neighbour analysis quantified the number of PD-1^+^ T cells nearby each PD-L1^+^ tumour cell within a specific radius; (**B**) proximity analysis quantified the distance of PD-1^+^ T cells from PD-L1^+^ tumour cells; (**C**) proximity analysis, closer view; (**D**) infiltration analysis quantified the density of PD-1^+^ T cells in 5 distance bands within and outside the tumour margin; (**E**) infiltration analysis, distance bands visualised on the real image; (**F**) density heatmap showing PD-L1^+^ tumour cells distributed mainly on the tumour margin.

**Figure 5 cancers-14-05387-f005:**
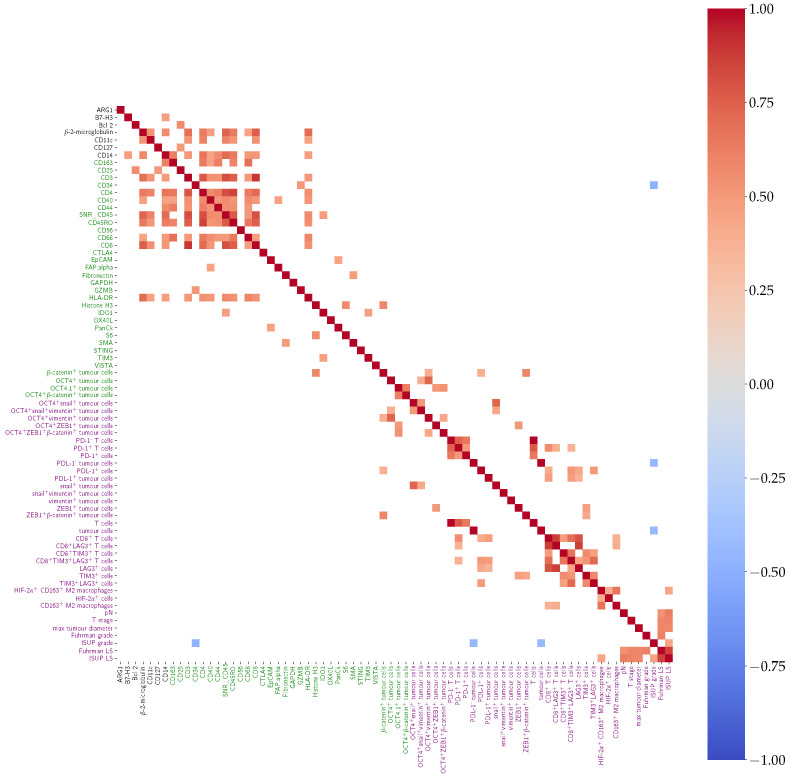
Heatmap showing positive (red), negative (blue), or no (white) correlation among molecular markers assessed with DSP (black), IF (green), and morphological (purple) features, aggregating the TMA core replicates by mean per patient. Only correlations with p<0.05 after Chow–Denning correction are shown.

**Figure 6 cancers-14-05387-f006:**
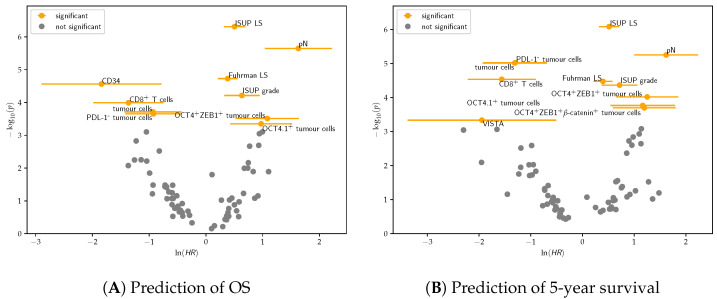
Volcano plots showing significant categorical univariate Cox regression features after Chow–Denning correction and their correlation with OS (**A**) and 5-year survival (**B**). Features with log(HR)>0 are positively correlated with the event, whereas those with log(HR)<0 are negatively correlated.

**Figure 7 cancers-14-05387-f007:**
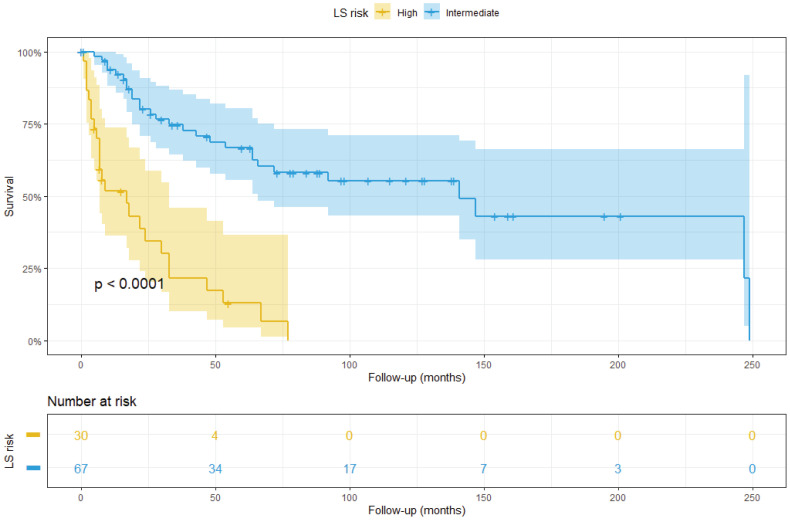
Kaplan–Meier plot showing patient at high (yellow) and intermediate (blue) risk according to ISUP LS.

**Figure 8 cancers-14-05387-f008:**
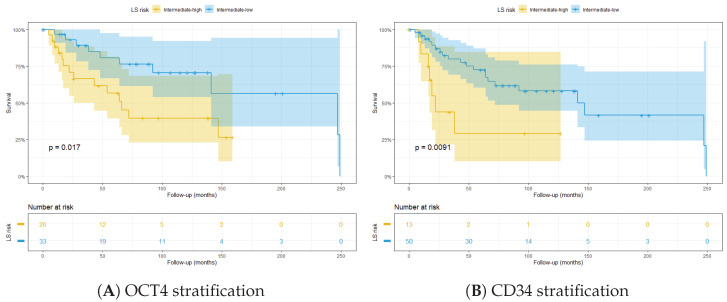
Stratification of patients at intermediate ISUP LS, based on: (**A**) OCT4 detected through multiplex IF; and (**B**) CD34 detected through DSP.

**Figure 9 cancers-14-05387-f009:**
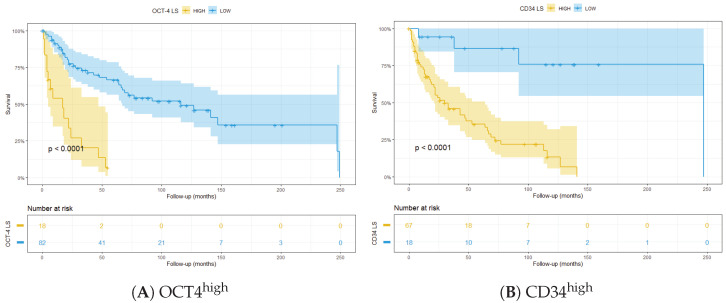
Kaplan–Meier plots showing stratification of patients with integrated LS using: (**A**) IF OCT4^high^; and (**B**) DSP CD34^high^.

**Figure 10 cancers-14-05387-f010:**
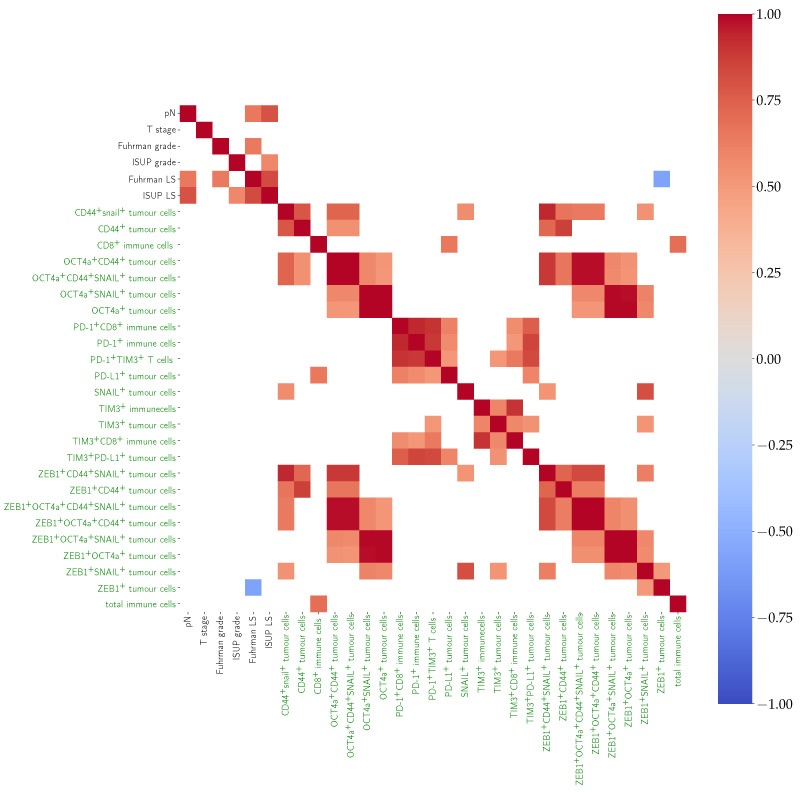
Heatmap showing positive (red), negative (blue), or no (white) correlation among morphological (black) and IF (green) markers in the WS cohort after unsupervised clustering. Features including OCT4a and ZEB1 showed the most correlation, while the only negative correlation was found between the Fuhrman LS and ZEB1^+^ tumour cells.

**Figure 11 cancers-14-05387-f011:**
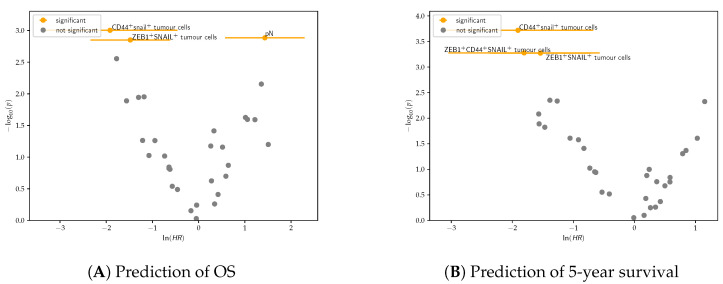
Volcano plots showing significant categorical univariate Cox regression features after Chow–Denning correction and their correlation with (**A**) OS and (**B**) and 5-year survival in the WS cohort. Features with log(HR)>0 are positively correlated with the event, whereas those with log(HR)<0 are negatively correlated.

**Figure 12 cancers-14-05387-f012:**
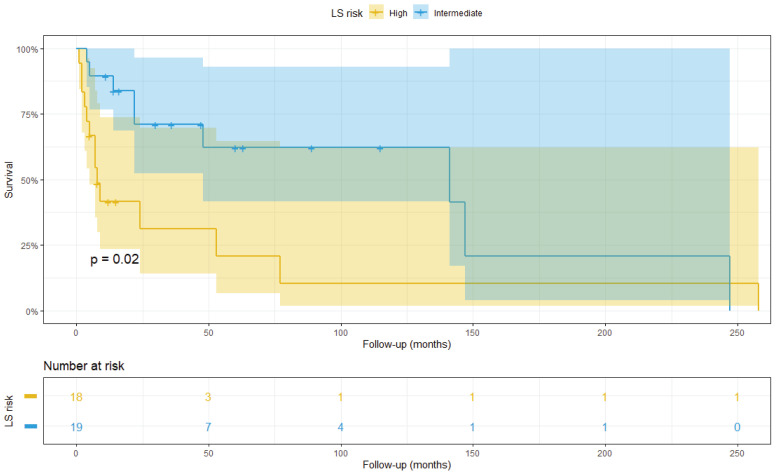
Patients belonging to the whole slide cohort stratified at high (yellow) and intermediate (blue) risk according to the ISUP LS.

**Table 1 cancers-14-05387-t001:** Table of pairs of markers that were insignificant in univariate Cox regression individually (p>0.05) but when used together as input to a multivariate Cox model achieved p<0.05 after Chow–Denning correction. Markers annotated with ↑ exhibit positive correlation with the event, whereas ↓ denotes negative correlation.

Survival	Markers		*p*-Value
5yr	↓ HLA-DR	↓ CD8^+^LAG3^+^ T cells	0.011
5yr	↓ ARG1	↑ OCT4^+^ tumour cells	0.016
5yr	↓ TIM3	↑ OCT4^+^β-catenin^+^ tumour cells	0.020
5yr	↓ ARG1	↑ OCT4^+^vimentin^+^ tumour cells	0.027
5yr	↑ OCT4^+^vimentin^+^ tumour cells	↓ snail^+^vimentin^+^ tumour cells	0.030
5yr	↓ HLA-DR	↑ OCT4^+^vimentin^+^ tumour cells	0.038
5yr	↓ CD8^+^TIM3^+^ T cells	↑ TIM3^+^ cells	0.044
overall	↑ OCT4^+^vimentin^+^ tumour cells	↓ PD-1^+^ cells	0.011
overall	↑ OCT4^+^ZEB1^+^β-catenin^+^ tumour cells	↓ snail^+^vimentin^+^ tumour cells	0.019
overall	↓ TIM3	↑ OCT4^+^β-catenin^+^ tumour cells	0.022
overall	↑ OCT4^+^vimentin^+^ tumour cells	↓ snail^+^vimentin^+^ tumour cells	0.022
overall	↑ OCT4^+^vimentin^+^ tumour cells	↓ snail^+^ tumour cells	0.035
overall	↓ PD-1^+^ cells	↓ snail^+^vimentin^+^ tumour cells	0.037

## Data Availability

Data are currently in the process of being made available, pending ethical approval.

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
