# Peer review of "Use of High-Plex Data Reveals Novel Insights into the Tumour Microenvironment of Clear Cell Renal Cell Carcinoma"

_cancers, 2022, doi:10.3390/cancers14215387_

Round 1
Reviewer 1 Report (New Reviewer)
This study was investigated the new potential biomarker for predicting the oncological outcomes in patients with ccRCC who underwent radical nephrectomy. Overall, this paper is well written. The reviewer would like to suggest some critiques as follows.
1. On line 11, “ICIs have significantly improved …(ccRCC) prognosis,” is unclear. “significantly improved the oncological outcomes” is better.
2. On line 11, 20, capital letters, including ccRCC, CSC, and EMT, were needed.
3. On line 31, what is “the deadliest genitourinary disease”? The authors should revise clearly.
4. On line 37, what is “drops to 12%”?
5. On line 93, “Understanding the high ccRCC … achieve personalized therapy” is unclear. “Intra-tumor heterogeneity” is important for this sentence. The authors should revise this sentence.
Author Response
Dear Reviewer,
Thank you very much for your precious feedback.
We amended the manuscript as suggested in your comments. In particular, we edited the sentence cited in the last point as follows:
Understanding how the high ccRCC ITH affects tumour progression and drug resistance is crucial for stratifying patients' prognosis and personalize the therapy.
We hope that now the manuscript is clearer.
Best wishes,
Raffaele
Reviewer 2 Report (New Reviewer)
In this manuscript, Filippis et al., systematically developed a semi-automated method to investigate 62 markers and their combinations in 150 primary ccRCCs using multiplex Immunofluorescence, NanoString GeoMx® Digital Spatial Profiling (DSP) and Artificial Intelligence (AI)-assisted image analysis in order to find novel prognostic signatures and investigate their spatial relationship. They found that coexpression of Cancer Stem Cell and Epithelial-to-Mesenchymal Transition markers such as OCT4 and ZEB1 are indicative of poor outcome. OCT4 and the immune markers CD8, CD34 and CD163 significantly stratified patients at intermediate LS. Furthermore, augmenting the LS with OCT4 and CD34 improved patient stratification by outcome. Their results support the hypothesis that combining molecular markers has prognostic value and can be integrated with morphological features to improve risk stratification and personalised therapy. To conclude, GeoMx® DSP and AI image analysis are complementary tools providing high multiplexing capability required to investigate the TME of ccRCC, while reducing observer bias.
1. Table 1 has a mistake. In Lymph node involvement, all scores are zero. The scores of pN1 and pN2 should be “2”.
2. Leibovich score is not the original contribution of this manuscript. References about Leibovich score should be identified in the title or remarks of Table 1.
3. In Figure12, please confirm whether the blue color represents the intermediate or low risk group.
Minor
1. In line 81, “Wnt/b-catenin” should be “Wnt/β-catenin’.
2. Scale bar is missing in panel D of Figure2. That would be clearer for readers if in the legend of panel D remarks what the gray represents.
3. In table A and line 120, “stage” should be replaced with “TNM stage” .
4. In table B2, the Oligonucleotide-conjugated antibodies consisted of five panels (immune cell profling, immune activation status, immune cell typing, tumour markers and drug targets). So it is recommended to sort according to five panels.
Author Response
Dear reviewer,
Thank you very much for your precious feedback.
We edited the manuscript as indicated in your comments. In particular, in figure 12 we confirmed that patients at intermediate risk are indicated in blue. Moreover, in figure 2 we explained that the gray dots are red blood cells which were excluded from the study, and a scale bar has been added.
In table A, "stage" refers to the pT stage. This has been specified in the table.
We hope that now the manuscript is clearer.
Best wishes,
This manuscript is a resubmission of an earlier submission. The following is a list of the peer review reports and author responses from that submission.
Round 1
Reviewer 1 Report
- In the introduction, the author mentioned that current prognostic tools are unable to accurately predict patient outcome. Please describe it more clearly for reader understanding.
- The tissue were collected in a wide range from 1983 to 2018. In this period, target therapy and immunotherapy had significant progression in prolonging survival. If possible, the data should be documented and discuss possible bias.
- The last paragraph might end with a brief conclusion of the study finding rather than limitation.
Author Response
Dear reviewer,
Thank you for you precious comments. Below you can find our point-by-point responses.
- In the introduction, the author mentioned that current prognostic tools are unable to accurately predict patient outcome. Please describe it more clearly for reader understanding.
We have explained how current prognostic tools only consider morphological tumour features and how these features are currently scored manually. These are the main limitations for an accurate prediction of outcome.
- The tissue were collected in a wide range from 1983 to 2018. In this period, target therapy and immunotherapy had significant progression in prolonging survival. If possible, the data should be documented and discuss possible bias.
None of these patients received treatment prior to surgery, and none received immune blockade therapy. Patient specimens were collected prior to therapy administration. This information has been added in the materials and methods section.
- The last paragraph might end with a brief conclusion of the study finding rather than limitation.
The section that includes the main findings has been moved to the end as a brief conclusion.
We hope these answer helped to clarify your doubts.
Best wishes,
Raffaele De Filippis
Reviewer 2 Report
Filippus et al. present a multiplex biomarker study based on nanostring technology using TMA and whole slide images. Their findings represent potential advances in digital pathology techniques, and they demonstrate prognostic significance for their work. However, the manuscript is not well constructed and lacks contextualization with the literature. Though there is likely significant value in the work, in its current state it is not ready for consideration.
Issues to Consider:
-The research is presented without sufficient context in terms of cited literature.
-The Discussion section is insufficient. It does not discuss the results in detail or provide suitable context in terms of the literature. Only 3 citations are given in the entire Discussion.
-The Figure quality is poor throughout and most would seem not to be worthy of presentation in the main text.
-It is unclear whether or not the work would be reproducible - it would seem the information given would be insufficient for this purpose.
-*Statistical methods used may require review by a biostatistician to determine suitability.
-The Results section is presented with subheadings and in the order expected of a Methods section. This along with many of the issues described above leads to a lack of narrative structure, which makes an already complex topic even more difficult to digest.
-The Figure legends are not sufficiently descriptive.
-Some portions of the text read more like an advertisement or informational pamphlet. This is related to the unqualified use of adjectives:
"Combining GeoMx® DSP with mIF is a great strategy to select specific ROIs and..."
"This is extremely important when considering the relationship..."
Author Response
Dear reviewer,
Thank you for you precious comments. Below you can find our point-by-point responses.
- The research is presented without sufficient context in terms of cited literature.
- The Discussion section is insufficient. It does not discuss the results in detail or provide suitable context in terms of the literature. Only 3 citations are given in the entire Discussion.
More literature has been cited in the discussion and we have sought to increase the contextual background of the study.
- The Figure quality is poor throughout and most would seem not to be worthy of presentation in the main text.
Figures were uploaded as .png files which correspond to the highest quality. All features in figure 1 are clearly readable if the page is zoomed: Figure 1 is important to give an idea about the large number of features which have been analysed, and the amount data which has been produced. Figure 2 shows the accuracy of tumour-stroma segmentation and cell segmentation, especially in ccRCC, where no standard tumour marker is used and where cell morphology and heterogeneity make tumour cells difficult to highlight by image analysis software. Moreover, in figure 2d one can appreciate how in Halo AI it was possible to profile different cell types only by looking at the cell morphology. Figure 3 gives a schematic representation of how immune cells were quantified at different distances from the tumour margin. Naming the distance bands (from band 1 to 9) was necessary to have a clear understanding of the spatial features showed in figure 1. Figure 4 shows how spatial analyses identified PD-1+ immune cells and PD-L1+ tumour cells mainly at the tumour border, indicating that spatial analysis is important to fully understand the role of immune cell infiltration in ccRCC. Like figure 1, figure 5 gives an overview of the molecular and morphological features assessed as long as their relationship. IN particular, it shows how DSP features were mainly correlated to each other, and that there was mainly a positive correlation. In our opinion, the volcano plots were the best way to summarise a such high number of cox regressions and gives a clear overview about features prognostic significance after p-value correction. Otherwise, a Kaplan Meier plot should have been shown for each significant feature. The reason why some Kaplan Meier plots have been included was to show LS accuracy in TMA and WS cohorts and how molecular features improved patient stratification.
- It is unclear whether or not the work would be reproducible - it would seem the information given would be insufficient for this purpose.
There are two aspects to reproducibility. The first, in terms of clinical significance would require a different test set which is out with the scope of this study. The second, technical reproducibility was not a focus of the study, but Nanostring has shown that the approach is reliable and reproducible in many different settings.
Reference: Validation of Antibody Panels for High-Plex Immunohistochemistry Applications. Douglas Hinerfeld, Kristi Barker, Chris Merritt, and Joseph Beechem. (2019) J Biomol Tech, 30(Suppl): S40–S41.
Reference: NanoString nCounter Technology: High-Throughput RNA Validation. Angela Goytain & Tony Ng. (2019) Chimeric RNA pp 125–139.
- Statistical methods used may require review by a biostatistician to determine suitability.
All raw data and statistical methods have been suggested and reviewed by a data analyst at Nanostring Technologies.
- The Results section is presented with subheadings and in the order expected of a Methods section. This along with many of the issues described above leads to a lack of narrative structure, which makes an already complex topic even more difficult to digest.
We have added more explanations and described the context of the study and we hope that the narrative flows better. We used headings on the Results section for clarity since we are attempting to incrementally build the use of digital pathology and have introduced new technologies as well as analyses.
- The Figure legends are not sufficiently descriptive.
Figure legends have been more fully described.
- Some portions of the text read more like an advertisement or informational pamphlet. This is related to the unqualified use of adjectives:
"Combining GeoMx® DSP with mIF is a great strategy to select specific ROIs and..."
"This is extremely important when considering the relationship..."
Agreed. Improperly used adjectives have been removed.
We hope these answer helped to clarify your doubts.
Best wishes,
Raffaele De Filippis
Reviewer 3 Report
Dear Authors,
thank you for sending this manuscript for evaluation.
Please consider:
- describe the experimental group in more detail; apart from gender, mean age etc. no clinical data are available. They are essential for interpretation of your research. This information + ccRCC status is not enough. TNM status, grade etc. Were all patients nephrectomized or did some receive NSS? How many were treated because of dissemination? Were they prospectively followed? How was data regarding OS status and CRD status gathered?
- Operative technique and nephrectomy qualification changed considerably over time. In my opinion population from '80 and second decade of XXI century is not the same. I have doubts regarding population homogenicity and influence of modern therapies on OS especially starting from 2000s. Please comment on that
- please consider adding presence or lack of necrosis on histopathology as one of established markers of ccRCC survival (as it is used in LS)
- why pT was not used? Only stage and tumor diamater
- please provide definition of ISUP LS
- for which proportion of patients was LS used? it is suitable for disseminated patients only. How was it calculated for whole population?
- please add commentary plus literature regarding how reliable a tissue that is FFPE for >30 years is
Author Response
Dear reviewer,
Thank you for you precious comments. Below you can find our point-by-point responses
- describe the experimental group in more detail; apart from gender, mean age etc. no clinical data are available. They are essential for interpretation of your research. This information + ccRCC status is not enough. TNM status, grade etc. Were all patients nephrectomized or did some receive NSS? How many were treated because of dissemination? Were they prospectively followed? How was data regarding OS status and CRD status gathered?
Information about TNM and grade has been added. In the Materials and Methods section it has been specified that this was a retrospective study and that all patients underwent total or partial nephrectomy prior to therapy administration, and none received immune blockade therapy. OS and CRD data were collected from the Research Tissue Bank of the Royal Infirmary of Edinburgh. This information has been added to the manuscript
- Operative technique and nephrectomy qualification changed considerably over time. In my opinion population from '80 and second decade of XXI century is not the same. I have doubts regarding population homogenicity and influence of modern therapies on OS especially starting from 2000s. Please comment on that
Possible issues regarding the large timespan of the population have been added in the discussion. A major aim of the study was to apply a multi-dimensional phenotyping approach to a cohort of patients in a clinical setting. We agree that conclusions about prognosis should be very cautious in this cohort, and must be further tested and validated, but we have shown the potential utility of this approach which is even more significant in the era of immune blockade therapy.
- please consider adding presence or lack of necrosis on histopathology as one of established markers of ccRCC survival (as it is used in LS)
Unfortunately, no necrosis information was present in our metadata as a single component of the LS, although LS scores were available. This has been clarified in the Materials and Methods section.
- Why pT was not used? Only stage and tumor diameter
If this is referring to LS, then we can confirm that it included pT, pN, pM, along with tumour diameter, presence of distant metastasis and necrosis. This has been clarified in the Materials and Methods section.
- Please provide definition of ISUP LS
ISUP LS stands for the Leibovich score in which the ISUP grading system was used instead of the outdated Fuhrman nuclear grade. A table describing the LS has also been added.
- For which proportion of patients was LS used? it is suitable for disseminated patients only. How was it calculated for whole population?
LS was applied to all patients with RCC and this is standard practice in many clinical settings.
- Please add commentary plus literature regarding how reliable a tissue that is FFPE for >30 years is
Problems relative to FFPE tissue block age has been added to the discussion. We have previously published validated mRNA extraction and analysis from similar old FFPE material.
Reference: Validation of a molecular and pathological model for five-year mortality risk in patients with early-stage lung adenocarcinoma. Raphael Bueno, Elisha Hughes, Susanne Wagner, Alexander S Gutin, Jerry S Lanchbury, Yifan Zheng, Michael A Archer, Corinne Gustafson, Joshua T Jones, Kristen Rushton, Jennifer Saam, Edward Kim, Massimo Barberis, Ignacio Wistuba, Richard J Wenstrup, William A Wallace, Anne-Renee Hartman, David J Harrison. (2015) Journal of Thoracic Oncology 10 (1), 67-73
We hope these answer helped to clarify your doubts.
Best wishes,
Raffaele De Filippis
Round 2
Reviewer 2 Report
The authors express in the response that the study is not reproducible due to lack of an additional test cohort.
Although modest improvements have been made to the manuscript, it still lacks a clear narrative structure and appears to be somewhat of a fishing expedition. If the overall hypothesis is that molecular markers measured by NS technology can improve prognostic stratification compared to LS alone, then more head-to-head comparisons should have been made - for instance - using ROC.
Much of the Results data remains of supplemental value to the main figures (2,4,6,7-9,11-12) and a schematic probably should have been included to clarify the focus and direction of the study.
The Results section remains a list of statistical tests and is not arranged to highlight the scientific findings even after revision. The readability in general is poor because of this and other issues mentioned above.
Author Response
Dear reviewer,
Thank you again for your feedback.
The aim of this study was to develop a methodology able to systematically explore kidney tumour microenvironment and demonstrate the feasibility of combined high-plexed labelling techniques and artificial intelligence in FFPE tissue sections. This method may be used in the future to have a general picture of the tumour molecular milieu and formulate novel hypotheses for more targeted experiments. The preliminary results presented in this study are a demonstration of this feasibility, although they need further confirmation.
The overall hypothesis is that high-plexed labelling techniques combined to machine-learning-based image analysis software are able to accurately explore molecular mechanisms in the tumour microenvironment. This approach extracts large amount of data while reducing inter-observer bias. Prognostically relevant molecular markers may then be used to improve current prognostic tools, such as Leibovich Score. This has been clarified in the text.
Narrative structure and readability have been improved in the results and discussion sections, and confusing/redundant numbers have been removed or collected in supplementary tables.
We hope this helped have a better understanding of the study.
Best wishes,
Raffaele De Filippis